# The urban built environment and adult BMI, obesity, and diabetes in Latin American cities

Cecilia Anza-Ramirez [1] ✉, Mariana Lazo[2,3], Jessica Hanae Zafra-Tanaka[1], Ione Avila-Palencia[2,4], Usama Bilal [2,5], Akram Hernández-Vásquez[1], Carolyn Knoll[2], Nancy Lopez-Olmedo[6], Mónica Mazariegos[7], Kari Moore [2], Daniel A. Rodriguez[8], Olga L. Sarmiento[9], Dalia Stern [10], Natalia Tumas [11,12,13] & J. Jaime Miranda [1,14] ✉

Latin America is the world's most urbanized region and its heterogeneous urban development may impact chronic diseases. Here, we evaluated the association of built environment characteristics at the sub-city −intersection density, greenness, and population density− and city-level −fragmentation and isolation− with body mass index (BMI), obesity, and type 2 diabetes (T2D). Data from 93,280 (BMI and obesity) and 122,211 individuals (T2D) was analysed across 10 countries. Living in areas with higher intersection density was positively associated with BMI and obesity, whereas living in more fragmented and greener areas were negatively associated. T2D was positively associated with intersection density, but negatively associated with greenness and population density. The rapid urban expansion experienced by Latin America provides unique insights and vastly expand opportunities for population-wide urban interventions aimed at reducing obesity and T2D burden.

Latin America is the world's most urbanized region, even surpassing China, which currently has ~60% urban population, 20 percentage points lower than Latin America[1]. Its urban population has rapidly increased from 40% in the 1950s to almost 80% in 2014[2], partly explained by a sustained rural-to-urban internal migration to cities driven by socioeconomic and political changes[3,4]. Thus, Latin America hosts several megacities with >10 million inhabitants and a large number of rapidly growing small and middle-sized cities. Such cities

have wide within- and between-city heterogeneity in their human-modified spaces, or built environments[5], including features such as density of street intersections, greenness, interrupted or fragmented urban development, isolation of that development, and population density[6–12].

Much of the scientific progress towards improved health has focused on the therapeutic aspects of disease treatment, but medical care is not necessarily the most powerful determinant of health

[1]CRONICAS Center of Excellence in Chronic Diseases, Universidad Peruana Cayetano Heredia, Lima, Peru. [2]Urban Health Collaborative, Dornsife School of Public Health, Drexel University, Philadelphia, PA, USA. [3]Department of Community Health and Prevention, Dornsife School of Public Health, Drexel University, Philadelphia, PA, USA. [4]Centre for Public Health, School of Medicine, Dentistry and Biomedical Sciences, Queen's University Belfast, Belfast, Northern Ireland, UK. [5]Department of Epidemiology and Biostatistics, Dornsife School of Public Health, Drexel University, Philadelphia, PA, USA. [6]Center for Population and Health Research, National Institute of Public Health, Cuernavaca, Mexico. [7]INCAP Research Center for the Prevention of Chronic Diseases (CIIPEC), Institute of Nutrition of Central America and Panama (INCAP), Guatemala City, Guatemala. [8]Department of City and Regional Planning, University of California, Berkeley, CA, USA. [9]School of Medicine, Universidad de los Andes, Bogota, Colombia. [10]CONACyT- Center for Population and Health Research, National Institute of Public Health, Cuernavaca, Mexico. [11]Department of Political and Social Sciences, Research Group on Health Inequalities, Environment, Employment Conditions Knowledge Network (GREDS-EMCONET), Universitat Pompeu Fabra, Barcelona, Spain. [12]Johns Hopkins University - Pompeu Fabra University Public Policy Center (UPF-BSM), Universitat Pompeu Fabra, Barcelona, Spain. [13]Centro de Investigaciones y Estudios sobre Cultura y Sociedad, Consejo Nacional de Investigaciones Científicas y Técnicas (CONICET) y Universidad Nacional de Córdoba, Córdoba, Argentina. [14]School of Medicine, Universidad Peruana Cayetano Heredia, Lima, Peru. ✉e-mail: cecilia.anza@upch.pe; jaime.miranda@upch.pe

outcomes, with environmental and social factors accounting for 50–60% of health gains[13–15]. This raises the question as to whether within- and between-city heterogeneity and other built environmental attributes are related to different health profiles in the Latin American region and similar areas undergoing major population and urbanization transitions. Among the health profiles of interest, we prioritize those related to chronic non-communicable conditions, which are rising around the globe and more so in resource-constrained countries[16–19].

Obesity and type 2 diabetes (T2D) are two leading chronic non-communicable conditions and major public health problems given their disability and mortality burdens[20]. The study of the built environment and its role in terms of cardiometabolic risk factors such as obesity and T2D in a rapidly changing region such as Latin America has not been studied across the range of cities or countries. Here, building upon previously reported relations with metabolic health outcomes in other settings[6–12], we studied the association between body mass index (BMI), obesity, and T2D, using individual data harmonized from health surveys from Latin American countries with built environment characteristics at the sub-city level (intersection density, greenness, and population density) and city level (fragmentation and isolation).

## Results

### City and population characteristics

We included a total of 93,280 survey respondents living in 675 sub-city units clustered in 233 cities for obesity and 122,211 survey respondents in 740 sub-city units clustered in 236 cities for T2D (Fig. 1 and Supplementary Table 1). In Brazil, Colombia, and Mexico, the reduction in the sample size (42% for anthropometry and 52% for diabetes) was due to BMI and T2D only being measured in a random subsample of respondents. Only 8% and 4% of participants in the anthropometry and diabetes modules, respectively, had missing data for these outcomes (Supplementary Table 1). Additional characteristics by country are provided in Supplementary Table 2.

Table 1 shows participant characteristics according to obesity and T2D status. Overall, the average age was 42 years, ~58% of the sample was females, and half of the sample completed high school education

or more. Respondents with obesity, as well as with T2D, tended to be older and have lower educational levels when compared to those without obesity or T2D, respectively (Table 1). Our characterization of the study sample by population density is available in Supplementary Table 3.

### Built environment characteristics

We found strong associations in opposite directions depending on the built environment attribute, and the direction of some of our findings do not align with those previously reported in the literature, mostly derived from high-income settings, signaling therefore to the particularities of the swift development and urbanization experienced by Latin America. All estimates are provided in Table 2, and visually in Fig. 2. Sensitivity analysis did not show major changes in all outcomes assessed (Supplementary Tables 4a–c).

### Higher intersection density and higher levels of BMI and obesity

Intersection density ($n/km^2$) measures the number of intersections per $km^2$ and represents street connectivity, which has previously been associated with a higher likelihood of walking and physical activity[6,7]. Our results show the higher the intersection density the higher the levels of BMI (0.1 units of BMI higher) or odds of obesity (4–5% higher odds) and diabetes (4% higher odds of diabetes). Whilst single exposure models (Model 1) show no evidence of an association with the outcomes studied, the inclusion of population educational attainment at sub-city level and percentage of the urban area (Model 2) provided clearer positive associations with BMI and obesity but not for diabetes. These associations were somewhat attenuated with the addition of the other built environment features studied (Model 3).

### Greenness is inversely associated with BMI and obesity (and diabetes)

Level of greenness, measured as Normalized Difference Vegetation Index (NDVI), has been associated with lower odds of central and peripheral obesity[8], adiposity[9], and T2D[10]. The exposure to areas with higher levels of greenness in Latin American cities show lower levels of BMI and lower odds of obesity and diabetes (Model 1), and these

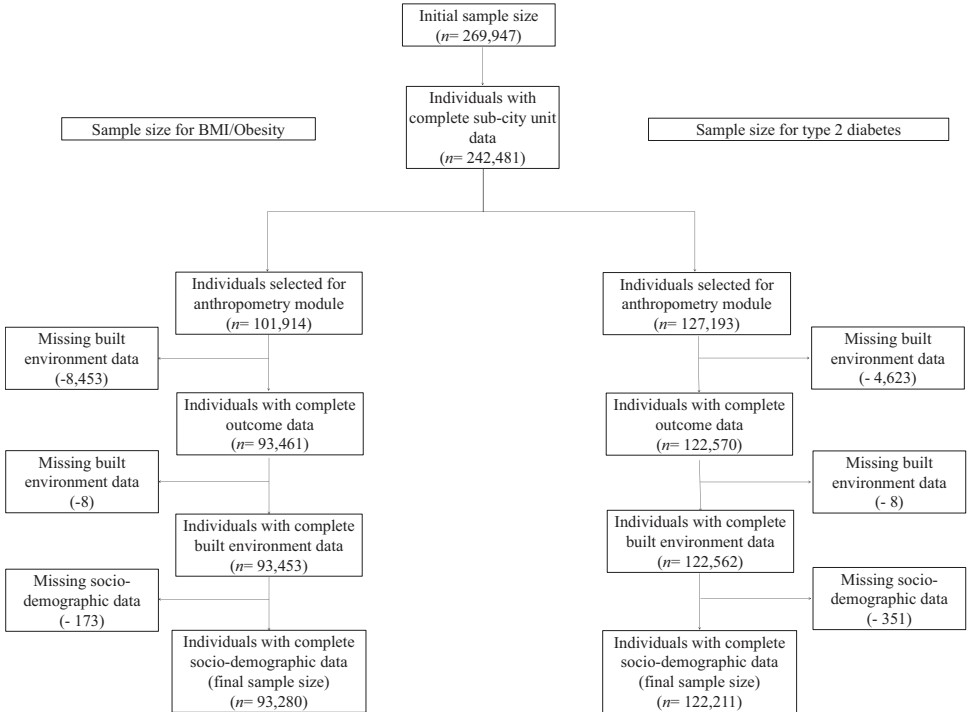

**Fig. 1 | Flowchart.** Flowchart describing the sample size and those with complete data analyzed for each of the outcomes.

**Table 1 | Characteristics of the study sample by obesity and type 2 diabetes status**

| Variables | Obesity | | Overall sample (N = 93,280) | Type 2 diabetes | | Over all sample (N = 122,211) |
|---|---|---|---|---|---|---|
| | No. [n = 69,724] (74.8%) | Yes [n = 23,556] (25.2%) | | No [n = 113,680] (93.0%) | Yes [n = 8,531] (7.0%) | |
| **Individual-level characteristics** | | | | | | |
| Age | 41.4 ± 17.1 | 45.3 ± 15.1 | 42.3 ± 16.7 | 41.5 ± 16 | 57.4 ± 14.6 | 42.6 ± 16.5 |
| Female sex | 56.2% | 62.5% | 57.8% | 58.4% | 60.4% | 58.5% |
| **Highest educational level completed** | | | | | | |
| Less than primary | 15.3% | 19.7% | 16.4% | 15.1% | 31.6% | 16.2% |
| Primary | 32.1% | 38.2% | 33.6% | 33.8% | 36.2% | 33.9% |
| Secondary | 38.2% | 31.4% | 36.5% | 37.3% | 22.2% | 36.3% |
| University or higher | 14.4% | 10.6% | 13.4% | 13.9% | 10.0% | 13.6% |
| **Sub-city level characteristics** | | | | | | |
| Intersection density ($n/km^2$) | 30 ± 36.1 | 29.7 ± 37.1 | 29.9 ± 36.3 | 29.3 ± 35.5 | 30.2 ± 35.7 | 29.4 ± 35.6 |
| Greenness (median NDVI) | 0.6 ± 0.2 | 0.6 ± 0.2 | 0.6 ± 0.2 | 0.6 ± 0.2 | 0.6 ± 0.2 | 0.6 ± 0.2 |
| Population density in built-up areas ($n/km^2$) | 7,862 ± 4,467 | 7,509 ± 4,245 | 7,773 ± 4,414 | 8,298 ± 4,903 | 7,530 ± 4,437 | 8,244 ± 4,876 |
| Population educational attainment (Z score) | 0.5 ± 1.4 | 0.3 ± 1.4 | 0.4 ± 1.4 | 0.5 ± 1.6 | 0.4 ± 1.5 | 0.5 ± 1.6 |
| **City-level characteristics** | | | | | | |
| Fragmentation [patch density ($n/100$ ha)] | 0.5 ± 0.3 | 0.4 ± 0.3 | 0.4 ± 0.3 | 0.5 ± 0.3 | 0.5 ± 0.3 | 0.5 ± 0.3 |
| Isolation [mean distance to the nearest urban patch within the geographic boundary(m)] | 83.2 ± 33.0 | 85.1 ± 35.0 | 83.7 ± 33.6 | 83.8 ± 35.7 | 84.6 ± 35.3 | 83.8 ± 35.7 |
| Percentage of urban area (%) | 8.8 ± 8.2 | 8.6 ± 8.1 | 8.8 ± 8.1 | 8.4 ± 7.6 | 8.7 ± 8 | 8.4 ± 7.6 |

NDVI normalized difference vegetation index. Values are show as percentage (%) or mean ± standard deviation.

observations became more pronounced for BMI (0.9 BMI units lower) and obesity (5% lower odds of obesity) but became attenuated for diabetes (Model 2). The inclusion of additional built environment variables in the models yielded estimates closer to the null (Model 3).

**Population density is negatively associated with diabetes only**
Population density has been linked with higher walking rates for commuting and for other purposes[11]. All single-exposure models for the three outcomes of interest show null associations with population density (Model 1). Further additional adjustments do not show major changes in the estimates for levels of BMI or odds of obesity, yet they strengthened for diabetes, yielding 4% lower odds of diabetes with higher population density (Model 3).

**City fragmentation is negatively associated with BMI and obesity**
The fragmentation of urban development of the city was measured using patch density. Patches describe contiguous urban development areas with large patches representing larger continuous areas and small patches representing small islands of discontinuous or leapfrog development. For a given area, more patches imply higher patch density and more fragmented development. Urban sprawl, a hallmark of fragmented development, is associated with lower walking and higher obesity prevalence[12]. We found that a higher urban fragmentation was associated with lower BMI (up to 0.16 units of BMI lower) or obesity (7% lower odds of obesity), albeit the strength of the evidence is borderline. There was no evidence of an association between city fragmentation and diabetes.

**City isolation is not associated with BMI, obesity, or diabetes**
The area-weighted mean distance to the nearest patch, usually accompanied by not accessible or affordable mass transit infrastructure, represents isolation. This characteristic may affect how much people need to travel to get to other places in the city, potentially reducing the

viability of active transportation and increasing obesity and T2D. As with city fragmentation, there was no evidence of an association between-city isolation and diabetes. On the contrary, city isolation was borderline associated with BMI (0.09 units of BMI higher) and obesity (3% higher odds of obesity) (Model 1), but further adjustment attenuated this association towards the null (Model 2 and Model 3).

## Discussion
In this study, leveraging a multinational sample in more than 200 Latin American cities, we examined associations between sub-city and city-built environment characteristics with BMI, obesity, and diabetes. Contrary to what has been reported in other settings, where a higher density of intersections in cities would bring benefits through increased walkability[12,21–25], in our sample of Latin American cities we found that higher intersection density was associated with higher BMI, and also with higher odds of having obesity and diabetes. Similarly, population density has been previously reported to be linked with increased active transportation and therefore lower obesity[11], and our findings do not support this observation as we found no association with BMI or obesity.

An ecological multi-country analysis has reported a positive link between an index of agglomeration as a proxy for a measure of urban concentration and a higher prevalence of T2D[26], but our results showed the opposite: the higher the population density, the lower the odds of T2D. Urban sprawl, a hallmark of fragmented development, is expected to be directly linked with higher obesity prevalence by reducing walking[12], yet in our Latin American cities we found the opposite, the higher the city fragmentation the lower the levels of BMI and odds of obesity. Consistent with the expected direction of associations previously reported in the literature, we found that the greener the areas in Latin America the better the cardiometabolic outcomes studied, and we also found a positive relationship between isolation in Latin American cities, i.e. longer distances to travel within cities, with BMI and obesity but not for diabetes.

**Table 2 | Associations between sub-city and city exposures and BMI, obesity, and type 2 diabetes**

| Sub-city and city characteristics (z scores) | Exposure contrast (SD)[†] | Model 1 | Model 2 | Model 3 |
|---|---|---|---|---|
| **Body mass index (n = 93,280)[‡]** | | Coefficient [95% CI] | Coefficient [95% CI] | Coefficient [95% CI] |
| Sub-city intersection density (n/km²) | 37.1 | 0.03 [−0.03; 0.10] | 0.10 [0.05; 0.16] | 0.10 [0.02; 0.19] |
| Sub-city greenness (median NDVI) | 0.2 | −0.06 [−0.17; 0.05] | −0.09 [−0.18; 0.00] | −0.04 [−0.15; 0.06] |
| Sub-city population density in built-up areas (n/km²) | 4,414 | −0.02 [−0.14; 0.10] | 0.02 [−0.09; 0.14] | −0.03 [−0.14; 0.08] |
| City fragmentation [patch density (n/100 ha)][§] | 0.3 | −0.15 [−0.31; 0.01] | −0.16 [−0.32; 0.00] | −0.13 [−0.29; 0.04] |
| City isolation [mean distance to the nearest urban patch within the geographic boundary(m)] | 33.6 | 0.09 [−0.01; 0.19] | 0.06 [−0.05; 0.17] | 0.04 [−0.07; 0.16] |
| **Obesity (n = 93,280)[‡]** | | OR [95% CI] | OR [95% CI] | OR [95% CI] |
| Sub-city intersection density (n/km²) | 37.1 | 1.02 [0.99; 1.05] | 1.05 [1.02; 1.09] | 1.04 [1.00; 1.08] |
| 10Sub-city greenness (median NDVI) | 0.2 | 0.96 [0.92; 1.00] | 0.95 [0.91; 0.99] | 0.97 [0.93; 1.01] |
| Sub-city population density in built-up areas (n/km²) | 4,414 | 1.00 [0.96; 1.05] | 1.02 [0.97; 1.07] | 1.00 [0.95; 1.04] |
| City fragmentation [patch density (n/100 ha)][§] | 0.3 | 0.94 [0.87; 1.01] | 0.93 [0.87; 1.00] | 0.95 [0.88; 1.02] |
| City isolation [mean distance to the nearest urban patch within the geographic boundary(m)] | 33.6 | 1.03 [0.99; 1.07] | 1.02 [0.97; 1.06] | 1.01 [0.96; 1.06] |
| **Type 2 diabetes (n = 122,211)** | | OR [95% CI] | OR [95% CI] | OR [95% CI] |
| Sub-city intersection density (n/km²) | 35.6 | 1.04 [0.99; 1.08] | 1.04 [0.98; 1.09] | 1.04 [0.98; 1.10] |
| Sub-city greenness (median NDVI) | 0.2 | 0.96 [0.93; 1.00] | 0.97 [0.93; 1.01] | 0.98 [0.94; 1.02] |
| Sub-city population density in built-up areas (n/km²) | 4876 | 0.99 [0.95; 1.04] | 0.99 [0.94; 1.03] | 0.96 [0.92; 1.00] |
| City fragmentation [patch density (n/100 ha)][§] | 0.3 | 0.97 [0.91; 1.04] | 0.97 [0.91; 1.04] | 0.98 [0.92; 1.04] |
| City isolation [mean distance to the nearest urban patch within the geographic boundary(m)] | 35.7 | 1.00 [0.96; 1.03] | 1.00 [0.96; 1.04] | 0.99 [0.96; 1.03] |

Coefficient or Odds Ratio (OR) with 95% confidence intervals (95% CI) are presented.

*NDVI* normalized difference vegetation index.

Model 1: single built environment exposure, adjusted for age, sex, education, and the country as a fixed effect.

Model 2: single built environment exposure adjusted for age, sex, education, population educational attainment at sub-city level, percentage of urban area, and country as a fixed effect.

Model 3: all built environment exposures, adjusted for age, sex, education, population educational attainment at sub-city level, percentage of urban area, and country as a fixed effect.

[†]All exposures were scaled based on the mean and standard deviation (SD) and therefore the unit of contrast is the SD.

[‡]Additionally, adjusted for age-squared.

[§]Models 1 adjusted for percentage of urban area.

Our study is unique in researching the built environment in a region in rapid transition, more so since Latin America has served as an urban natural experiment to the globe due to its large population and city changes that have occurred over a brief period relative to other regions. Although selection bias resulting from residents clustering in different parts of a city due to, for example, the geographical accessibility to a number of services, is a weakness of other studies, the way we defined cities as a collection of smaller municipalities in this study increased the heterogeneity within the spectrum of urban contexts and thus minimizes some of these concerns.

The rapid enlargement of urban areas in Latin America does not necessarily resemble the same pattern of the suburbanization phenomenon experienced in North America, where those who have higher income tend to relocate away from urban areas. In our characterization of the study sample by population density, we observed that less populated areas have also lower percentages of urban areas, and in terms of our outcomes of interest, obesity, and diabetes, there is a gradient with higher proportions of these two conditions in the less populated areas, compared to the most densely populated areas.

Our statistical handling is also informative as we present estimations based on single and multiple adjustments to inform the magnitude and direction of the association of any given built environment attribute evaluated, ranging from 0.09–0.16 less BMI units and 3% to 7% lower odds of obesity or T2D. Given the challenges with the units used for the reporting of weight loss interventions[27], the reductions in BMI found in our study are somewhat in line with the magnitude of effects derived from weight loss interventions with a diversity of programs and intensities observed in clinical trials[28–30]. From another angle, statistical simulations of a projected population-wide reduction of 1% in BMI values would substantially avoid up to 2.1–2.4 million incident cases of T2D over 20 years[31]. These findings are of considerable policy relevance as they complement scientific therapeutical progress made to date, and vastly expand actionable opportunities to design and implement population-based urban interventions to reduce the burden of obesity and diabetes by targeting built environment characteristics. As such, our study results could inform future decision-making scenarios targeting certain built environment attributes, particularly in those cities that have not yet achieved their full growth in the region.

The rationale for the selection of obesity and diabetes as outcomes of interest lies in their key role as cardiometabolic diseases, their rapid rise over the years in low- and middle-income countries, and therefore the major population health burden associated with these conditions[32–34]. Although there is no conclusive evidence about the causal pathway through which obesity may lead to T2D[35], obesity is known to occur many years before the initial signs or diagnosis of T2D, being one of the modifiable factors that could be intervened in order to prevent T2D[28]. Healthy diets and physical activity could contribute to reducing the risk of these two chronic non-communicable conditions[36,37], but it is also known that these two factors worsen with urbanization[26,38].

The introduction of sub-city and city-scale policies focusing on the urban environment exposures should be considered given the potential to reduce obesity and T2D[39]. Our results yield estimates that are not directly comparable with pharmacological treatment effects, however, it is well known that reductions in weight could delay the progression towards diabetes[40,41]. Given the population-wide nature of the exposures studied, modest effects can have wide-reaching repercussions on the burden of obesity and T2D in large populations, as shown recently in the case of blood pressure and hypertension[42,43]. A modest population-wide reduction in BMI was associated with very strong declines in T2D incidence and mortality in Cuba[44]. The rationale

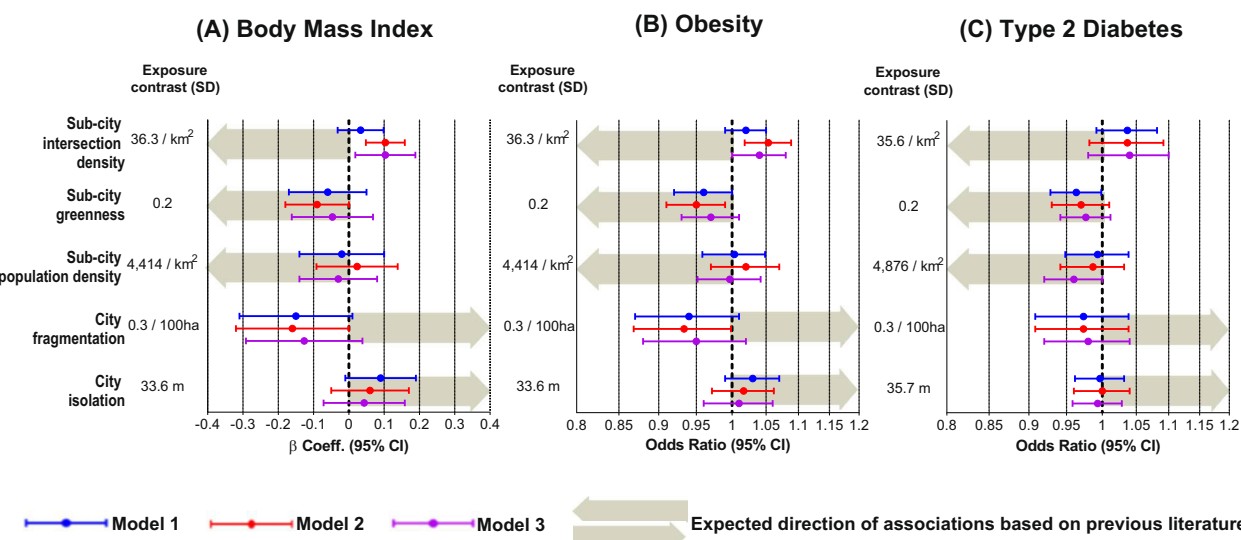

**Fig. 2 | Associations between built environment characteristics, as single (each built environment exposure at a time) or multiple exposure models (multiple built environment exposures together), and BMI, obesity, and type 2 diabetes, adjusted for covariates.** All exposures were standardized to a mean of zero and a standard deviation (SD) of 1, therefore the exposure contrast used is 1 SD for each variable. For body mass index (BMI, $n = 93,280$), the β coefficient represents the mean change in BMI per one SD increase in the exposure, adjusted for covariates. For obesity ($n = 93,280$) and type 2 diabetes (T2D, $n = 122,211$), each odds ratio (OR) represents the relative change in the odds of having obesity or having T2D per one SD increase in the exposure, adjusted for covariates. Error bars show 95% confidence intervals (95% CI). Arrows indicate the expected direction of the associations based on previous literature (further information about these previously reported associations is provided in the Results, Discussion, and Methods sections of the manuscript). Arrows pointing to the left indicate a negative association, that is, the higher the exposure, the lower the BMI, the lower the odds of having obesity, or the lower the odds of having T2D. Conversely, arrows pointing to the right indicate a positive association, that is, the higher the exposure, the higher the BMI, the higher the odds of having obesity, or the higher the odds of having T2D. Model 1: single built environment exposure, adjusted for age, sex, education, and the country as a fixed effect. Model 2: single built environment exposure adjusted for age, sex, education, population educational attainment at sub-city level, percentage of urban area, and country as a fixed effect. Model 3: all built environment exposures, adjusted for age, sex, education, population educational attainment at sub-city level, percentage of urban area, and country as a fixed effect. For BMI and obesity, all models are additionally adjusted for age-squared. For city fragmentation, Models 1 are adjusted for percentage of urban area. X axis in obesity and T2D are in logarithmic scale.

behind shifting entire population distributions is accompanied by the elevated individual and societal costs brought by obesity, T2D, and their related complications[20]. Albeit small, the effect sizes of the observed associations are important as they are directly related to characteristics influencing the health determinants in large population groups[18,45–47].

Different studies indicate that ensuring walkability is an important feature to promote physical activity, which has a positive effect on metabolic outcomes[22–28]. In that regard, increased local intersection density, higher greenness, and decreased city fragmentation and isolation may improve active transportation[8,9,12,21,23,25,48–50]. Previous studies from different parts of the world, yet many of them originating in high-income countries, have reported that living in high-walkable areas characterized by higher intersection density, greener areas, lower development isolation, and lower development fragmentation[6–12] was associated with lower BMI, and a lower likelihood of obesity and T2D[9,21,51–54]. However, contrary to what has been reported in other settings[23,25], we found that higher BMI and higher odds of obesity were observed in sub-city units with higher intersection density and lower fragmentation. Our study also signals the complexity of engaging in physical activity in urban areas of Latin American which may be difficult due to a variety of urban-related factors[55,56], including, but not limited to, the perception of safety and violence[57,58].

Fragmentation has been associated with an increase in transportation infrastructure, such as large roads, which may act as mobility barriers for pedestrians and as an automobile-oriented transport planning of the city[59], therefore reducing physical activity due to a more sedentary transportation mode through vehicle ownership. In Latin America, our findings of higher fragmentation being associated with lower BMI and obesity could reflect a reliance on the use of public

transportation[60], and therefore maintaining some degree of physical activity related to active commuting. Due to the increasing urbanization and expansion that is still occurring in Latin America[61], the land use and design may be different from those of upper middle- and high-income countries, and increased intersection density may not necessarily mean safety or pedestrian-friendly paths. By way of contrast, for example, in the U.S., it has been found that higher intersection density was associated with higher food store density suggesting a better availability of healthy food[62]. However, as shown in Brazil, a higher concentration of stores is associated with higher food consumption[63], thus, we hypothesize that the proliferation of convenience stores as well as unhealthy street food, may explain why intersection density is associated with higher BMI and higher odds of obesity in Latin American cities.

Some limitations merit attention. First, for some characteristics, the sub-city level approach might be too large to appropriately capture the relevant spatial context. Studies with smaller areas, 'true' neighborhoods, e.g., census tracts, may find different results than ours and be more aligned with prior literature on this topic; however, accessing data at a smaller level might be difficult for all the Latin American cities studied here. Second, the outcome data was derived from surveys implemented between 2002 and 2017, and the built environment characteristics data for the years 2010 and 2018, indicate a temporal mismatch. However, prior research has suggested that the built environment changes slowly over time[64], and our sensitivity analyses, using surveys administered in the year 2010 or later, did not show substantively different results. Additionally, we adjusted for a country-level fixed effect in our models to account for potential unmeasured differences between countries. We also adjusted for features related to the survey's sampling design such as age, sex, and sub-city

socioeconomic status. Third, given the cross-sectional design, we cannot ascertain causality. In this type of analysis, isolating the effects of built environment characteristics on health outcomes of longer onset may be difficult given the cumulative effects of several environmental attributes as well as that of some socioeconomic factors that were accounted for in this analysis. Accordingly, unmeasured and residual confounding is likely to remain. In addition, the use of and proximity to green areas, perceptions of safety and crime, sedentary lifestyle, food environment, geographical accessibility to health care services, among others, are some variables that could well be defined as mediators[65] in the associations reported in this manuscript, highlighting the complexities in the relationship between the urban environment and individual outcomes.

Despite these concerns, there are significant strengths. This is a multi-country study, potentially the first study of its kind in Latin America, based on a unique data set involving 10 countries and a large number of sub-city units and cities. Additionally, the multilevel approach and multiple exposure analyses allowed a comprehensive understanding of the variability and influence of the built environment on BMI, obesity, and T2D in the region. Considering that modifications of the built environment can have direct impacts in promoting better profiles of physical activity and enhancing the quality of food systems to the entire population[66], these results may be the basis to inform future studies in which experimental or quasi-experimental changes in the built environment exert an impact on the outcomes studied here, an effect that is mediated through changes in patterns of physical activity and food systems.

In this study of Latin American cities, sub-city intersection density, greenness, population density, and city fragmentation were positively or negatively associated with BMI, obesity, and T2D, and our findings in the built environment attributes of intersection, population density and fragmentation do not align to what has been previously described in the literature, signaling the challenges of a rapid urban expansion and growth as experienced by the Latin American region. Our results have important implications for designing and implementing urban interventions aimed at reducing obesity and T2D burden.

## Methods

### Study setting

The SALURBAL project (Salud Urbana en America Latina/Urban Health in Latin America)[67] compiled and harmonized social, built environment and health survey data to examine multilevel aspects of health across all cities with ≥100,000 inhabitants as of 2010 in 11 Latin American countries (Argentina, Brazil, Chile, Colombia, Costa Rica, Guatemala, Mexico, Nicaragua, Panama, Peru, and El Salvador)[67]. A city was defined as the combination of administrative units (i.e., sub-city units) that overlap with the urban extent of the city, as determined by satellite imagery. Thus, a city may include a single administrative unit or a combination of adjacent administrative units[67]. Using this approach, 371 cities were identified and operationalized as clusters of the smallest administrative units (1436 sub-city units, most commonly named *municipios, comunas or distritos*). The SALURBAL data resource includes several harmonized health survey data (non-communicable disease risk factors, adult and children) and geolinked to vast built, natural, and social environment data for up to three levels: cities, sub-city units (e.g., *Municipios* of counties that compose cities), and neighborhoods (similar to US census tracts).

In this cross-sectional multilevel analysis, we focused on adults 18 years old and older, living in sub-city units and cities for which we were able to identify and link selected social and built environment data with individual-level T2D, obesity, and demographic information. The analytical data set comprised data from the following nationally representative health surveys, with years given in brackets: Argentina (2013), Brazil (2013), Chile (2010), Colombia (2007), Guatemala (2002),

Mexico (2012), Nicaragua (2003), Panama (2007), Peru (2016), and El Salvador (2004) (Fig. 1). Supplementary Table 5 provides details of the health surveys included in the study, Supplementary Table 6 outlines the information of variables included at the individual, sub-city, and city level, and Quistberg et al.[67] details specific data sources.

### Outcomes

**BMI and obesity.** BMI was computed as weight (kg)/height$^2$ (m). Except for Argentina, weight and height were measured by trained personnel using standard protocols[67]. If more than one measurement was taken, an average of all available measures was calculated. For Argentina, self-reported data for weight and height were available and included in the analysis with the other countries. We used BMI as a continuous variable and also defined obesity as a BMI ≥ 30 kg/m$^2$ that was compared with people with BMI < 30 kg/m$^2$.

**Type 2 diabetes.** For all countries, T2D was defined as a self-reported physician diagnosis of high blood sugar levels or diabetes. Although gestational diabetes confers an increased lifetime risk of T2D, the questionnaires did not clearly distinguish between gestational diabetes and later progression to T2D. As such, given that our research question was focused on the built environment as it relates to T2D, we excluded women who answered yes to a gestational diabetes diagnosis[68]. For Panama, the original survey questions did not allow separating gestational diabetes.

### Exposures

Exposures of interest were built environment characteristics at the sub-city level (intersection density, greenness, and population density) and city level (fragmentation and isolation), detailed in Supplementary Table 6. These built environment attributes were purposely selected to include variables with previously reported associations with metabolic health outcomes[6–12], and these variables are linked with the promotion of active lifestyles through walking, or other physical activity-enhancing behaviors[6,69,70]. Supplementary Table 6 describes data sources, year, definition, and interpretation of these exposures.

### Covariates

We identified a number of potential variables of adjustment at the individual level: age, sex, and maximum educational level completed (less than primary, primary, secondary, or university); at the sub-city level: population educational attainment (a summary score of sub-city education); and at the city level: percentage of urban area (proportion of the city that is covered by built-up/urban patches). For fragmentation, the percentage of urban area is used as an adjustment variable to account for some surrounding non-built-up areas sub-city units, which could introduce measurement error due to the heterogeneity in urbanized areas. See Supplementary Table 6 for details.

### Statistical analysis

We used descriptive statistics of individual, sub-city, and city-level variables to characterize the sample by obesity and T2D status. The supplementary material contains basic descriptive statistics by country and by population density (Supplementary Tables 2 and 3). Additionally, we evaluated the correlations between exposure variables using Pearson (Supplementary Table 7). The highest correlation was seen between-city percentage of urban area with city fragmentation ($r = 0.7$); however, the former variable is used for adjustment in order to account for some surrounding non-built-up areas sub-city units used in the fragmentation calculation, which could introduce measurement errors due to the heterogeneity in urbanized areas (Supplementary Table 6).

We assessed the association of sub-city and city-level built environment characteristics with BMI using linear multilevel

models, and with obesity and T2D using logistic multilevel models. In both cases, we included sub-city and city random intercepts, and fitted three types of models: model 1, single built environment exposure models adjusted for individual-level characteristics (age, sex, and educational level); model 2, model 1 further adjusted for sub-city population educational attainment, and city percentage of urban area; and model 3, includes all built environment exposures adjusted for variables described in model 2 (multi-exposure model). All models included country as a fixed-effect and were performed with complete case analysis for the obesity/BMI and T2D samples separately.

The built environment is linked to physical activity and thus indirectly to BMI[11]. For T2D models, we did not adjust for BMI since we consider BMI as an intermediate variable lying on the causal pathway between the built environment and T2D occurrence, Previous literature has suggested that adjusting for mediator factors may lead to flawed inferences[71].

For comparability purposes, exposures were standardized to a mean of zero and a standard deviation of 1. For BMI, the β coefficient represents the mean change in BMI per one standard deviation increase in the exposure, adjusted for covariates. For obesity and T2D, each odds ratio (OR) represents the relative change in the odds of having obesity or having T2D per one standard deviation increase in the exposure, adjusted for covariates. ORs are presented in the figures using a logarithmic scale ($X$ scale).

Last, we did not use sampling weights in our analysis[72,73] as we focused on studying the associations with the built environment characteristics at the sub-city and city level rather than reporting prevalence estimates. Furthermore, the survey weights included in the analytical data sets of the surveys are appropriate for their use when analyzing the entire national sample. SALURBAL harmonized survey data sets include only a subset of the population living in urban areas.

All analyses were conducted in SAS/STAT® software v9.4 (SAS Institute Inc., Cary, NC, USA). The code is available on GitHub at https://github.com/Drexel-UHC/SALURBAL-MS62.

### Sensitivity analyses
Sensitivity analyses were performed excluding Argentina for obesity and BMI, and Panama for T2D due to the difference in the definition of the outcomes. For obesity, additionally, sensitivity analysis comparing subjects with normal weight ($<25$ kg/m$^2$) vs. overweight and obesity ($\geq25$ kg/m$^2$) was performed. To see if different data acquisition time has an effect on our results, we explored a sensitivity analysis excluding surveys conducted before 2010. Additionally, extreme values in BMI and population density were detected and a sensitivity analysis excluding them was run. Given that our interest was in associations, and not in estimating prevalence, we opted not to use survey weights. However, some of the variables used in the computation of weights are included as covariates in the analysis (including age and sex).

### Ethics approval
The SALURBAL study protocol has been approved by the Drexel University Institutional Review Board with ID #1612005035.

### Reporting summary
Further information on research design is available in the Nature Portfolio Reporting Summary linked to this article.

## Data availability
The SALURBAL study obtained health survey data from health and/or statistical agencies within each country. Data from Brazil (https://www.ibge.gov.br/estatisticas/sociais/saude/9160-pesquisa-nacional-de-saude.html), Chile (http://epi.minsal.cl/encuesta-ens/), Mexico (https://ensanut.insp.mx/), Panama (http://www.gorgas.gob.pa/enscavi/), Peru (http://webinei.inei.gob.pe/anda_inei/index.php/catalog/563), and El Salvador (http://www.datos.gob.sv/dataset/encuesta-nacional-de-enfermedades-cronicas) are publicly available at the links provided. Data from Argentina, Colombia, Guatemala, and Nicaragua are available under restricted access due to data use agreements between the SALURBAL Study and statistical agencies within the country. Requests for the harmonized data set can be obtained by contacting the SALURBAL project salurbal.data@drexel.edu and after completing a data use agreements. Requests are reviewed by the Data Methods Core and Publications & Presentations Committee on a monthly basis. To learn more about SALURBAL's data set, visit https://drexel.edu/lac/ or contact the project at salurbal@drexel.edu.

## Code availability
The analytic code, including descriptive analysis and regression models,is provided at https://github.com/Drexel-UHC/SALURBAL-MS62.

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

## Acknowledgements
The authors acknowledge the contribution of all SALURBAL project team members. For more information on SALURBAL and to see a full list of investigators, see https://drexel.edu/lac/salurbal/team/. The SALURBAL project (*Salud Urbana en América Latina*, Urban Health in Latin America) is funded by the Wellcome Trust [205177/Z/16/Z]. More information about the project can be found at www.lacurbanhealth.org. UB was supported by the Office of the Director of the National Institutes of Health under award number DP5OD26429. NT was supported by the European Union's Horizon 2020 research and innovation program under the Marie Sklodowska-Curie grant agreement No. 89102. The funding sources had no role in the analysis, writing, or decision to submit the manuscript. Please visit https://drexel.edu/lac/data-evidence for a complete list of data sources. The authors would like to thank Jacqueline Anne Seiglie for her valuable comments in earlier drafts.

## Author contributions
Conceptualization: C.A.-R., J.Z.-T., I.A.-P., U.B., A.H., C.K., N.L.-O., M.M., K.M., D.A.R., O.L.S., D.S., N.T., J.J.M. Funding acquisition: D.A.R., O.L.S., J.J.M. Methodology: C.A.-R., M.L., J.Z.-T., U.B., J.J.M. Data curation: M.L., K.M.. Formal analysis: C.A.-R., M.L., C.K. Visualization: C.A.-R. Supervision: ML, J.J.M. Writing—original draft: C.A.-R., J.J.M. Writing—review & editing: C.A.-R., M.L., J.Z.-T., I.A.-P., U.B., A.H., C.K., N.L.-O., M.M., K.M., D.A.R., O.L.S., D.S., N.T., J.J.M.

## Competing interests
The authors declare no competing interests.
