## [Peer Review File · Nature Communications]

Reviewers' Comments:

Reviewer #1:

Remarks to the Author:

All comments we raised in a previous round of review have been addressed and we feel that this manuscript is now acceptable for publication.

Reviewer #2:

Remarks to the Author:

I appreciate the efforts in responding to the previous review, and your clarification on the focus on the built environment.

That is an interesting exercise reproducing models that have been used in another region to Latin America.

Many of the methodological questions regarding clarification on the data and the plausibility of some of the assumptions used in the model have not been fully addressed in the answers nor in the manuscript.

Reviewer #3:

Remarks to the Author:

I have gone through the paper, the comments from the referees, and the response from the authors. This is the first time that I have read the paper.

The paper addresses an important issue that might be of interest to a broad international audience.

Due to the multinational nature of the research question, a cross-sectional design might be the most feasible. However, such a design has several limitations that should have been more clearly addressed in the paper. The major limitation in a cross-sectional design is the inability to determine the temporal sequence of exposure and outcome.

Moreover, it should be clear that prevalence is a function of the incidence and duration of the disease. Several factors have impact on the prevalence.

The following factors increase the prevalence:

1. Longer disease duration
2. Improved survival
3. Increase incidence
4. In-migration of cases
5. In-migration of susceptible individuals
6. Out-migration of healthy people
7. Improved diagnostic abilities

The following factors decrease the prevalence:

1. Shorter disease duration
2. High case-fatality rate
3. Decrease incidence
4. In-migration of healthy people

5. Out-migration of cases

6. Improved cure rate of cases (which in this study is not really a major problem for diabetes since it is a chronic disease)

It should also be emphasized what the odds ratios estimate in a cross-sectional study, namely the prevalence odds ratio (and not risk ratio as mentioned on page 11 of the response letter).

Moreover, there is an assumption for the calculation of odds ratio, namely the rare disease assumption.

The authors should address if there is any potential selection bias in the surveys (the reader does not get much information on the participation rate) and the impact on the estimates.

Nor does the reader get any information on comorbidities. Obesity is a strong risk factor, not only for type 2 diabetes, but also for cardiovascular diseases and some major cancer types.

Moreover, there are several complications to type 2 diabetes such as cardiovascular, eye, and kidney diseases.

In many countries the severity of diseases may affect where you live due to access to medical care and thus the prevalence estimates.

Therefore, both potential selection bias and residual confounding have impact on the prevalence estimates. This is only briefly covered in the paper and the response letter.

A minor comment: the authors describe data by a mean and standard deviation in several places in the study. It is clear, for example from Table 2, that many of the variables do not follow a normal distribution since some of the participants will have negative values.

The authors should avoid focusing on statistical significance but only look at the estimates. Please see Amrhein V et al. Scientists rise up against statistical significance. *Nature*. 2019; 567:305-307.

Manuscript Reference: NCOMMS-21-42602-T

Manuscript Title: The urban built environment and adult BMI, obesity and diabetes in Latin American cities: A cross-sectional multilevel analysis using individual and contextual-level data

REVIEWER COMMENTS

Reviewer #1

All comments we raised in a previous round of review have been addressed and we feel that this manuscript is now acceptable for publication.

Response: Thank you for your time, we appreciate this response.

Reviewer #2

I appreciate the efforts in responding to the previous review, and your clarification on the focus on the built environment.

Response: Many thanks for the positive feedback.

That is an interesting exercise reproducing models that have been used in another region to Latin America.

Response: We appreciate the feedback, recognising the value of our work focusing on the Latin American region.

Many of the methodological questions regarding clarification on the data and the plausibility of some of the assumptions used in the model have not been fully addressed in the answers nor in the manuscript.

Response: We would like to seek editorial guidance on the specific aspects requiring further attention.

Reviewer #3

I have gone through the paper, the comments from the referees, and the response from the authors. This is the first time that I have read the paper. The paper addresses an important issue that might be of interest to a broad international audience.

Response: Many thanks for the positive feedback.

Due to the multinational nature of the research question, a cross-sectional design might be the most feasible. However, such a design has several limitations that should have been more clearly addressed in the paper. The major limitation in a cross-sectional design is the inability to determine the temporal sequence of exposure and outcome.

Response: The reviewer is correct, and we concur that a cross-sectional design has limitations, which we initially reported in our discussion, and now have expanded (see underlined text).

“Given the cross-sectional design, we cannot ascertain causality due to the inability to determine the temporal sequence of exposure and outcome, and therefore this design allows an understanding of city-related factors that could explain the variability in obesity and T2DM across Latin American cities.”

Moreover, it should be clear that prevalence is a function of the incidence and duration of the disease. Several factors have impact on the prevalence.

The following factors increase the prevalence:

1. Longer disease duration
2. Improved survival
3. Increase incidence
4. In-migration of cases
5. In-migration of susceptible individuals
6. Out-migration of healthy people
7. Improved diagnostic abilities

The following factors decrease the prevalence:

1. Shorter disease duration
2. High case-fatality rate
3. Decrease incidence
4. In-migration of healthy people
5. Out-migration of cases
6. Improved cure rate of cases (which in this study is not really a major problem for diabetes since it is a chronic disease)

Response: We appreciate the constructive feedback provided. As per the previous response, we have edited our limitations. Additionally, we want to highlight that our aim is not to report the

prevalence of the conditions studied, but focus instead on the associations with built environment characteristics at the sub-city and city-level.

It should also be emphasized what the odds ratios estimate in a cross-sectional study, namely the prevalence odds ratio (and not risk ratio as mentioned on page 11 of the response letter). Moreover, there is an assumption for the calculation of odds ratio, namely the rare disease assumption.

Response: We agree with the reviewer, and we did not use RR in the manuscript. The rare disease assumption is required if the odds ratio is used to estimate a prevalence ratio (or risk ratio in the case of incident odds ratios). But regardless of whether the outcome is rare or not, the OR is a valid and interpretable measure of association.¹ We have carefully reviewed our paper to make sure that we are not incorrectly interpreted OR as prevalence ratios. Additionally, it is worth mentioning that the logistic regression represents a stable model (i.e. less issue of non-convergence).

The authors should address if there is any potential selection bias in the surveys (the reader does not get much information on the participation rate) and the impact on the estimates.

Response: We concur that this information is essential. One of our strengths relies on using nationally-representative surveys, which do not preclude selection biases but will likely minimise such bias. No changes were made.

Nor does the reader get any information on comorbidities. Obesity is a strong risk factor, not only for type 2 diabetes, but also for cardiovascular diseases and some major cancer types.

Response: We agree, ideally, we would like to have access to such nationwide information to have a more comprehensive assessment of the role of built environment on chronic conditions, including comorbidities. Unfortunately, many national surveys focus on few conditions, which preclude an extensive assessment of comorbidities. Indeed, access to cancer prevalence data carries its own challenges, also present in the Latin American setting,² which are beyond the scope of this manuscript.

Moreover, there are several complications to type 2 diabetes such as cardiovascular, eye, and kidney diseases.

Response: See response above, similar limitations apply to complications related to type 2 diabetes, but the information is not available.³

In many countries the severity of diseases may affect where you live due to access to medical care and thus the prevalence estimates.

Response: Correct, and this refers to geographical accessibility.⁴ Of note, in the Latin American region, there are also other barriers to access care, including socioeconomic factors and ability to pay and out-of-pocket expenditures.^{5,6}

Therefore, both potential selection bias and residual confounding have impact on the prevalence estimates. This is only briefly covered in the paper and the response letter.

Response: We appreciate these reflections. We have expanded on the issues of selection bias and residual confounding due to challenges with access to care as potential determinants of the prevalences observed in our study. The new text reads as follows:

Although selection bias resulting from residents with particular characteristics clustering in different parts of a city due to, for example, the geographical accessibility to health care services, is a weakness of other studies, the way we defined cities as a collection of smaller municipalities in this study minimizes these concerns.

In this type of analysis, isolating the effects of built environment characteristics on health outcomes of longer onset may be difficult given the cumulative effects of several environmental attributes as well as that of some socioeconomic factors not measure. Accordingly, other variables were not available, and therefore unmeasured and residual confounding is likely. The use of and proximity to green areas, perceptions of safety and crime, sedentary lifestyle, food environment, geographical accessibility to health care services, among others are some of these variables.

A minor comment: the authors describe data by a mean and standard deviation in several places in the study. It is clear, for example from Table 2, that many of the variables do not follow a normal distribution since some of the participants will have negative values.

Response: We thank the reviewer for mentioning this. The main variable that you may be referring to is population educational attainment, which is a Z-score, and thus it can have negative values. We have added “z score” next to the variable in the table and where appropriate.

The authors should avoid focusing on statistical significance but only look at the estimates. Please see Amrhein V et al. Scientists rise up against statistical significance. Nature. 2019; 567:305-307.

Response: We very much align with this recommendation, thank you. We have removed the term ‘statistical significance’ from our manuscript and rephrased the sentences where needed.

References

1. Schmidt, C. O. & Kohlmann, T. When to use the odds ratio or the relative risk? *Int. J. Public Health* **53**, 165–167 (2008).
2. Piñeros, M. *et al.* Progress, challenges and ways forward supporting cancer surveillance in Latin America. *Int. J. Cancer* **149**, 12–20 (2021).
3. Zhang, P. *et al.* Global epidemiology of diabetic foot ulceration: a systematic review and meta-analysis †. *Ann. Med.* **49**, 106–116 (2017).
4. Carrasco-Escobar, G., Manrique, E., Tello-Lizarraga, K. & Miranda, J. J. Travel Time to Health Facilities as a Marker of Geographical Accessibility Across Heterogeneous Land Coverage in Peru. *Front. Public Heal.* **8**, 498 (2020).
5. Correa-Burrows, P. Out-Of-Pocket Health Care Spending by the Chronically Ill in Chile. *Procedia Econ. Financ.* **1**, 88–97 (2012).
6. Emmerick, I. C. M., Luiza, V. L., Camacho, L. A. B., Vialle-Valentin, C. & Ross-Degnan, D. Barriers in household access to medicines for chronic conditions in three Latin American countries. *Int. J. Equity Health* **14**, 1–14 (2015).

Reviewers' Comments:

Reviewer #3:

Remarks to the Author:

I find that the authors have revised the paper in a satisfactory manner.

Reviewer #4:

None

Reviewer #5:

Remarks to the Author:

There is little information regarding the original data sources for the health survey data used in the SALURBAL data platform. The supplementary material only lists sample sizes. The cited paper by Quistberg does not contain information about the health surveys. The authors need to clarify the details of the health surveys used to build the population and provide health outcomes and individual-level data for this study. This can go in the supplement, but the name, sampling plan, spatial and temporal resolution and data collection method (interviewer, self-report, linkage to medical record, etc) needs to be provided for each country. In addition, please clarify that the years in parentheses are the year of the data used.

In addition, the authors need to acknowledge the limitations posed by harmonizing data from these very different sources. For example, harmonizing data from two different decades. This is not to say there aren't strengths of harmonization and pooling of data; there absolutely are. But, there are limitations when harmonizing across data using different sampling schemes, time periods etc that could affect results.

By not using the sampling weights and accounting for design, the surveys stop being nationally representative. I would edit the paper to make that clear. This is especially true if any of the surveys used over- or under-sampling in their schemes.

In addition, using a national survey does not remove the potential for selection bias, particularly when there is so much missing data. As suggested by prior reviewers, please acknowledge the limitation of selection bias induced by the large amount of missing data.

Many of the variables (e.g. food systems) mentioned in previous reviews could be correlated with built environment through economic development (i.e. not necessarily caused by built environment directly) and thus could represent potential confounders of the observed relationships. The authors slightly touch on this with their sentence about residual confounding. But, this should be expanded more in the discussion. The authors can't be sure that relationships observed are truly due to built environment or could be due to features correlated with built environment.

Manuscript title: “The urban built environment and adult BMI, obesity, and diabetes in Latin American cities: A cross-sectional multilevel analysis using individual and contextual-level data”

Reference: NCOMMS-21-42602C

Point-by-point response letter

We appreciate the reviewers’ comments and feedback on our manuscript. In this letter, we provide point-by-point responses to them in blue. Thanks, again, for your time and consideration.

Reviewer #3:

1. I find that the authors have revised the paper in a satisfactory manner.
Response: Thank you for your time, we really appreciate your input.

Reviewer #5:

1. There is little information regarding the original data sources for the health survey data used in the SALURBAL data platform. The supplementary material only lists sample sizes. The cited paper by Quistberg does not contain information about the health surveys. The authors need to clarify the details of the health surveys used to build the population and provide health outcomes and individual-level data for this study. This can go in the supplement, but the name, sampling plan, spatial and temporal resolution and data collection method (interviewer, self-report, linkage to medical record, etc) needs to be provided for each country. In addition, please clarify that the years in parentheses are the year of the data used.

Response: We concur that this information is important. We have added Supplemental Table 5 with the requested survey information. Additionally, the following text (highlighted) has been added in our manuscript’s first paragraph of Study Settings section in Methods:

“... Data from nationally representative health surveys, with years given in brackets, of the following countries was available: Argentina (2013), Brazil (2013), Chile (2010), Colombia (2007), Guatemala (2002), Mexico (2012), Nicaragua (2003), Panama (2007), Peru (2016), and El Salvador (2004). Supplementary Table 5 provides details of the health surveys included in the study, Supplementary

Table 6 outlines the information of variables included at the individual, sub-city, and city level, and Quistberg et al details specific data sources.”

2. In addition, the authors need to acknowledge the limitations posed by harmonizing data from these very different sources. For example, harmonizing data from two different decades. This is not to say there aren't strengths of harmonization and pooling of data; there absolutely are. But, there are limitations when harmonizing across data using different sampling schemes, time periods, etc., that could affect results.

Response: We agree that there are important limitations in effort related to data harmonization, and we have now expanded on the discussion's limitations of our manuscript:

“Second, the outcome data was derived from surveys implemented between 2002 and 2017, and the built environment characteristics data for the years 2010 and 2018, indicating a temporal mismatch. However, prior research has suggested that the built environment changes slowly over time,¹ and our sensitivity analyses, using surveys administered in the year 2010 or later, did not show substantively different results. Additionally, we adjusted for a country-level fixed effect in our models to account for potential unmeasured differences between countries. We also adjusted for features related to the survey's sampling design such as age, sex, and sub-city socioeconomic status.”

3. By not using the sampling weights and accounting for design, the surveys stop being nationally representative. I would edit the paper to make that clear. This is especially true if any of the surveys used over- or under-sampling in their schemes.

Response: We concur with this observation about nationally representativeness. As we are not intending to report prevalence estimates but associations, weights are not necessary.^{2,3} The following text has been added at the end of the first paragraph of Study Settings, in Methods:

“We focused on studying the associations with the built environment characteristics at the sub-city and city-level rather than reporting prevalence estimates, and therefore we did not use sampling weights in our analysis.^{2,3}”

4. In addition, using a national survey does not remove the potential for selection bias, particularly when there is so much missing data. As suggested by prior reviewers, please acknowledge the limitation of selection bias induced by the large amount of missing data.

Response: We would like to clarify that the drop in the number of observations is not due to missing data. For Brazil, Colombia, and Mexico, not all initial survey respondents were selected to complete all survey modules. These surveys implemented a random selection of adults to complete the anthropometry and/or health modules. A large proportion of what appears to be missing outcomes data was due to survey respondents not being selected for these modules, i.e. 42% for anthropometry, and 52% for diabetes. We have updated **Supplemental Table 1** to reflect this information. Among those selected for these modules, only 8% for anthropometry and 4% for diabetes, respectively, had missing outcome data.

Of the respondents who were selected for the diabetes module, those who had missing outcome data were older (43 years vs 38 years), male (57% vs 41%), and had lower education level (21% vs 16%) compared to those with outcome information. Of the respondents who were selected for the anthropometry module, those with missing outcome data were more frequently male (50% vs 42%) and had lower education level (21% vs 16%) than those without missing anthropometry outcome data, but these two groups did not differ by age (average was 42 years for both groups). (Data not shown). We concur that selection bias remains even in nationally representative surveys and, the demographic variables mentioned above were included as covariates in our regression analysis.

We have modified the manuscript accordingly. We have added the following text in the first paragraph of the Results section:

“We included a total of 93,280 survey respondents living in 675 sub-city units clustered in 233 cities for obesity and 122,211 survey respondents in 740 sub-city units clustered in 236 cities for T2DM (Figure 2 and Supplementary Table 1). In Brazil, Colombia, and Mexico, the reduction in the sample size (42% for anthropometry and 52% for diabetes) was due to BMI and T2DM only being measured in a random subsample of respondents. Only 8% and 4% of participants in the anthropometry and diabetes modules, respectively, had missing data for these outcomes (Supplementary Table 1).”

5. Many of the variables (e.g. food systems) mentioned in previous reviews could be correlated with built environment through economic development (i.e. not necessarily caused by built environment directly) and thus could represent potential confounders of the observed relationships. The authors slightly touch on this with their sentence about residual

confounding. But, this should be expanded more in the discussion. The authors can't be sure that relationships observed are truly due to built environment or could be due to features correlated with built environment.

Response: Thanks for pointing this out. Many variables could be confounders because of their relationship with the built environment and the outcome of interest, but also mediators. Confounding does not necessarily imply a causal relationship among the variables,⁴ and most of the variables that we have identified (green areas, perceptions of safety and crime, sedentary lifestyle, food environment, geographical accessibility to health care services) are more likely to operate as mediators because they lie on the causal pathway of our association of interest. Therefore, we feel much more comfortable using the term residual confounders for those unknown variables and we are using the term mediators for those that we have identified and fulfill the criteria to be defined as a mediator. We have added the following sentence (highlighted text) to the manuscript's discussion about the study limitations:

“Third, given the cross-sectional design, we cannot ascertain causality. In this type of analysis, isolating the effects of built environment characteristics on health outcomes of longer onset may be difficult given the cumulative effects of several environmental attributes as well as that of some socioeconomic factors that were accounted for in this analysis. Accordingly, unmeasured and residual confounding is likely to remain. In addition, the use of and proximity to green areas, perceptions of safety and crime, sedentary lifestyle, food environment, geographical accessibility to health care services, among others, are some variables that could well be defined as mediators⁴ in the associations reported in this manuscript, highlighting the complexities in the relationship between the urban environment and individual outcomes.”

References

1. Hirsch, J. A. *et al.* How much are built environments changing, and where?: Patterns of change by neighborhood sociodemographic characteristics across seven U.S. metropolitan areas. *Soc. Sci. Med.* **169**, 97–105 (2016).
2. Bollen, K. A., Biemer, P. P., Karr, A. F., Tueller, S. & Berzofsky, M. E. Are Survey Weights Needed? A Review of Diagnostic Tests in Regression Analysis. *Annual Review of Statistics and Its Application* vol. 3 375–392 (2016).
3. Avery, L. *et al.* Unweighted regression models perform better than weighted regression techniques for respondent-driven sampling data: Results from a simulation study. *BMC Med. Res. Methodol.* **19**, 1–13 (2019).
4. MacKinnon, D. P., Krull, J. L. & Lockwood, C. M. Equivalence of the mediation, confounding and suppression effect. *Prev. Sci.* **1**, 173–181 (2000).

Reviewers' Comments:

Reviewer #5:

Remarks to the Author:

The authors sufficiently addressed my comments from the prior review.